# Study of the Effects of the Structure of Phthalazinone’s Side-Group on the Properties of the Poly(phthalazinone ether ketone)s Resins

**DOI:** 10.3390/polym11050803

**Published:** 2019-05-05

**Authors:** Feng Bao, Fengfeng Zhang, Chenghao Wang, Yuanyuan Song, Nan Li, Jinyan Wang, Xigao Jian

**Affiliations:** 1State Key Laboratory of Fine Chemicals Dalian University of Technology, Dalian 116024, China; bfisvip@163.com (F.B.); Zhangfeng0908@126.com (F.Z.); wangchh@mail.dlut.edu.cn (C.W.); polymerlinan@dlut.edu.cn (N.L.); jian4616@dlut.edu.cn (X.J.); 2Department of Polymer Science and Materials, Dalian University of Technology, Dalian 116024, China; 3Shenzhen China Start Optoelectronic Technology Co., Ltd, Shenzhen 518132, China; yysong1211@163.com

**Keywords:** phthalazinone, side-group, thermal properties, processability

## Abstract

The application of poly(phthalazinone ether ketone)s (PPEKs) resin containing phthalazinone moiety is limited, due to its poor thermoforming processability. To investigate the effects of the phthalazinone’s side-group on the thermal stability and processability of the resin, a series of PPEKs resins with different side-group (–H/–CH_3_/–Ph) were prepared by nucleophilic aromatic substitution polymerization. The properties of the obtained resins were investigated by differential scanning calorimetry analysis (DSC), thermogravimetric analysis (TGA), dynamic thermomechanical analysis (DMA), and rheogoniometer. The results show that the introduction of methyl or phenyl into the PPEKs resin, significantly reduced the melting viscosity of the resin, but resulted in a slight decrease in the thermal stability of it. This might be due to the presence of methyl or phenyl, which enhanced the free volume of the molecule and reduced the entanglement between the chains; the results of the computer simulation confirmed it. Moreover, the resin films displayed excellent tensile strength with the introduction of methyl or phenyl. In a word, a novel poly(phthalazinone ether ketone)s resin with thermal resistance, easy processing and excellent mechanical properties could be obtained by introducing appropriate bulk-rigid side-groups into the phthalazinone moiety.

## 1. Introduction

The fiber reinforced plastic (FRP) materials have seen considerable growth in recent years, due to the growing demand for lightweight materials in the automobile, aerospace, and other key fields. The FRP materials can be divided into FRTP (fiber reinforced thermoplastic plastic) and FRSP (fiber reinforced thermoset plastic), according to the resin materials. Compared with FRSP materials, FRTP materials have attracted the attention of more researchers, due to the fact that that FRTP materials are more designable, have a shorter production period, and are easier to store. The properties of FRTP materials are determined by the properties of reinforcing materials and resin materials. Therefore, resin materials with excellent mechanical properties, superior thermal stability, and good processability, have been expected with the expansion of the application of FRTP materials, under harsh conditions.

Poly(phthalazinone ether ketone)s (PPEKs) [1,2,3,4] resins is a kind of high-performance engineering plastics, with phthalazinone moiety in its molecular chain. There is a consensus that PPEKs resin is considered as a satisfactory matrix resin material in the field of composite materials, owing to its attractive thermal stability, excellent solubility, and superior mechanical properties. Previous studies have revealed that the excellent thermal stability and solubility of PPEKs resins, are due to the fact that the phthalazinone moiety, which has totally aromatic, twisted, and noncoplanar structural features, enhance the rigidity of resin while destroying the regularity of resin [5,6]. Nevertheless, the twisting characteristics of the phthalazinone monomer lead to an increase in the entanglement of molecular chains and a significant decrease in the thermoforming processability, which is not conducive to the expansion of PPEKs resin applications, especially in the field of composite materials. Hence, improving the thermoforming processability, while maintaining the attractive thermal resistance, excellent solubility, and superior mechanical properties of PPEKs resin, is of great importance. Zhang [7] and Liao [8] effectively improved the thermoforming processability of PPEKs resin, by blending some resins with better thermoforming processability, such as Poly(ether sulfone) (PES) and thermotropic liquid crystals (TLCP). However, improvements of the thermoforming processability of the resin were accompanied by significantly decreased mechanical properties. Poor compatibility the between blending components is responsible for the reduced mechanical properties. Hence, Sun [9] tried to improve the thermoforming processability by inserting regular Poly(ether ether ketone ketone) (PEEKK) oligomers into PPEKs chains. However, this approach did not meet the satisfactory expectation due to the difficulty in obtaining high-molecular-mass PEEKK oligomers in solution polymerization systems [10]. Therefore, we tried to improve the thermoforming processability of the PPEKs resin, through random copolymerization, in a previous work. The results exhibited that copolymerization with some bulky monomers such as 9,9-bis(4-hydroxyphenyl) fluorene (BHPF) [11,12] and 2,5-bis(4-fluorobenzoyl) furan(BFBF) [13] can improve the thermoforming processability of PPEKs resin to some extent. Nevertheless, it is still important to improve the thermoforming processability of PPEKs, while maintaining the excellent thermal stability and mechanical properties of PPEKs.

The substantial entanglement between the PPEKs molecular chains is likely responsible for the poor thermoforming processability of PPEKs. Previous modification studies of PPEKs resin have focused on reducing the entanglement of molecular chains, but have rarely involved the 4-(4-Hydroxyphenyl) phthalazin-1(2*H*)-one (DHPZ) moiety. The purpose of this study was to investigate the effects of phthalazinone’s side-group on the properties, including thermal behavior, mechanical property, and thermoforming properties of PPEKs resin containing the phthalazinone moiety. Here, we have prepared three kinds of bisphenol monomers (DHPZ, DHPZ-M, DHPZ-Ph), whose side-groups of those monomers were H, methyl, and phenyl, containing phthalazinone moiety. Then, the poly(phthalazinone ether ketone)s (PPEK, PPEK-M, PPEK-Ph) were obtained by nucleophilic aromatic substitution polymerization. FTIR, NMR, WXRD, and GPC techniques were used to confirm the structure of these polymers. The thermal behavior and thermoforming properties of the resins was investigated by DSC, TGA, Dynamic mechanical analysis (DMA), and rotating rheometer. Additionally, the mechanical properties and solubility of the copolymers were also evaluated.

## 2. Materials and Methods

### 2.1. Materials

4-(4-Hydroxyphenyl) phthalazin-1(2*H*)-one (DHPZ, 99%) was kindly offered by Dalian Polymer New Materials Co., Ltd. (Dalian, China). Bis(4-fluorophenyl)-methanone (DFK) was obtained from Wuhan Yuancheng Technology Co., Ltd. (Wuhan, China). Phthalic anhydride (PA, 99%), *o*-cresol (98%), anhydrous aluminum chloride (AlCl_3_, 99%), and anhydrous potassium carbonate (K_2_CO_3_, 99%) were purchased from Aladdin Co., Ltd. (Shanghai, China) Anhydrous potassium carbonate (K_2_CO_3_, 99%) was ground and dried at 100 °C, for 24 h, under vacuum before use. Reagent-grade sulfolane (Aladdin, Shanghai, China) was refluxed over sodium hydroxide and then distilled under reduced pressure. Unless otherwise specified, the materials involved in the study, such as phenol, ethanol, and toluene were commercially available and used without further purification.

### 2.2. Characterization

Fourier-transform infrared (FTIR) spectra were measured on a NEXUS EURO, with a resolution of 4.0 cm^−1^ in the 4000–650 cm^−1^ region. The resin was prepared as a 1 mg·mL^−1^ solution and then coated on the potassium bromide pellet. NMR spectra were obtained on a Bruker DRX 400 NMR spectrometer (Bruker, Karlsruhe, Germany) with deuterated chloroform or DMSO as the solvent, and tetramethylsilane (TMS) as the internal standard. The molecular masses of the resins were carried out on a PL50 HPLC gel permeation chromatograph (GPC) instrument (Agilent Technologies, Santa Clara, CA, USA), with the flow phase of chloroform, and the monodisperse polystyrene as calibration. The chromatographic column was composed of two PLgel MIXED-C (5 m × 300 mm) columns. The morphological characteristics of resin were performed on a Bruker D8 wide-angle X-ray diffraction (WAXD) instrument (Bruker, Karlsruhe, Germany) using Cu-Kα radiation (wavelength λ = 0.15418 nm) of 5 and 80°, at a scanning rate of 10 °C·min^−1^. The glass transition temperature (*T_g_*) of the resin was recorded by a Mettler DSC822 (Mettler-Toledo, Zurich, Switzerland) differential scanning calorimeter. Approximately 5 mg of each sample was placed in a test tray, heated from 25 to 380 °C, at a rate of 10 °C·min^−1^ in a nitrogen atmosphere. The Mettler TGA/SDTA851 Thermogravimetric Analyzer (Mettler-Toledo, Zurich, Switzerland) was used to analyze the weight loss of the resin, when heated from 30 to 800 °C, with different heating rates (5, 10, 15, and 20 °C·min^−1^) under N_2_ or air atmosphere, at a flow rate of 50 mL·min^−1^. Rheological studies were measured by a TA ARX2000 instrument (TA Instruments, Newcastle, DE, USA), at a frequency of 1 Hz from 280 to 420 °C. Dynamic mechanical analysis (DMA) measurements were carried out with a DMA861e instrument (Mettler-Toledo, Zurich, Switzerland). Each film sample was cut into a 9 × 3 mm^2^ rectangle and then heated from 30 to 300 °C, at a heating rate of 5 °C·min^−1^. The tensile strength was determined using an Instron-5869 machine, with a capacity of 100 N, according to the ASTM D882-2018 standard.

### 2.3. Synthesis of Monomers

The monomers DHPZ-M and DHPZ-Ph were synthesized according to the literature [14,15,16]. The synthetic routes of DHPZ-M and DHPZ-Ph are shown in Scheme 1. The NMR, HRMs, and FTIR were used to verify the structure of the monomer, and the results are shown in Appendix A.

DHPZ-M—1H NMR (400MHz, DMSO-d6, TMS): δ=12.79 (s, 1H, 8-NH), 9.75 (s, 1H, 1-OH), 8.47 – 8.29 (m, 1H, 7-H), 7.98 – 7.83 (m, 2H, 5&6-H), 7.81 – 7.75 (m, 1H, 4-H), 7.33 (t, *J* = 6.6 Hz, 1H, 9-H), 7.28 (dd, *J* = 8.2, 2.2 Hz, 1H, 3-H), 7.02 (d, *J* = 8.2 Hz, 1H, 2-H), 2.27 (s, 3H, 10-H) 13C NMR (400 MHz, DMSO, TMS) δ = 159.6 (s, 12-C), 156.6 (s, 1-C), 147.1 (s, 6-C), 133.9 (s, 8-C), 132.1 (s, 13-C), 131.8 (s, 9-C), 129.7 (s, 11-C), 128.4 (s, 3&5-C), 127.3 (s, 7-C), 126.5 (s, 4-C), 126.1 (s, 10-C), 124.5 (s, 14-C), 114.9 (s, 2-C), 16.5 (s, 15-C).

DHPZ-Ph—1H NMR (400 MHz, DMSO-d6, TMS): δ = 12.84 (s, 1H, 8-H), 10.06 (s, 1H, 1-H), 8.37 (dd, *J* = 7.7, 1.3 Hz, 1H, 7-H), 7.98 ~ 7.79 (m, 3H, 5&6&9-H), 7.70 ~ 7.61 (m, 2H, 10-H), 7.50 (t, *J* = 5.3 Hz, 1H, 4-H), 7.48 ~ 7.40 (m, 3H, 3&11-H), 7.40 ~ 7.30 (m, 1H, 12-H), 7.18 (d, *J* = 8.3 Hz, 1H, 2-H). 13C NMR (400 MHz, DMSO, TMS) δ =159.7 (s, 12-C), 155.5 (s, 1-C), 146.8 (s, 6-C), 138.5 (s, 15-C), 134.0 (s, 13-C), 131.9 (s, 8-C), 131.8 (s, 7-C), 130.1 (s, 3-C), 129.7 (s, 11-C), 129.6 (s, 16-C), 128.5 (s, 17-C), 128.4 (s, 5-C), 128.2 (s, 14-C), 127.3 (s, 18-C), 127.2 (s, 9-C), 126.8 (s, 4-C), 126.5 (s, 10-C), 116.6 (s, 2-C).

### 2.4. Synthesis of Polymers PPEKs (PPEK, PPEK-M, and PPEK-Ph)

The PPEKs resins were successfully prepared by nucleophilic aromatic substitution polycondensation between DFK and 3a~3c. The synthetic route to PPEKs is shown in Scheme 2. All resins were obtained using a similar procedure, according to previous work [12,13]. The polymerization of PPEK-M is described as an example. DHPZ-M (23.5 mmol, 5.9284 g), DFK (23 mmol, 5.0187 g), potassium carbonate (32.9 mmol, 4.5469 g), 10 mL of sulfolane, and 10 mL of toluene were added to a dry, three-neck 100 mL round-bottom flask. The flask was equipped with a stirrer, a condenser, a Dean–Stark trap, and a nitrogen inlet. The reaction system was heated to 142 °C for 4 h in a thermostat-controlled oil bath, to accelerate the phenolate formation and distill the water. Later, the temperature of the reaction was raised to 165 °C, and the reaction was carried out for 2 h, to remove toluene completely. The reaction temperature was then elevated to 190 °C, until the reaction was completed. The reaction solution was poured slowly into hot water containing a small amount of hydrochloric acid with vigorous stirring. At this time, a cluster of fine white fibers was obtained. The acquired fiber was placed in boiling water, for more than 12 h, to remove the excess potassium carbonate. Then, the polymers were dried by placing the boiled fibers in a 120 °C oven, for 24 h. Finally, the purified products were obtained after drying in a 120 °C vacuum oven, for 24 h. In addition, some of the PPEKs resins were prepared into films, for the relevant test, and the preparation method was described in the literature [12].

## 3. Results

### 3.1. Polymer Synthesis

Our goal was to investigate the effect of the pendant group of the monomers containing the phthalazinone moiety on the properties of PPEK resin, including thermal behavior, mechanical property, and thermoforming properties. In order to minimize the effects of molecular mass on the properties of the resin, the molecular mass of the resin was controlled to be 20,000 g·mol^−1^ by adjusting the molar feeding ratio of monomer, while maintaining the reaction temperature and time consistence. The molecular masses of the PPEKs were measured by gel permeation chromatography (GPC). Table 1 presents the number-average molecular masses (*M_n_* values), weight-average molecular masses (*M_w_* values), and polydispersity indices (PDI values) of the PPEKs resin. The GPC traces are shown in Appendix A. Obviously, the *M_n_* values of the obtained resins are consistent with the design. This could help to reduce the effect of the molecular weight in a later discussion.

### 3.2. Structural Characterization

The chemical structures of the PPEKs resins were determined by ^1^H NMR and FTIR. Figure 1a shows the ^1^H NMR spectrum of the acquired resins. All signals were processed correctly and corresponded perfectly to the protons of the corresponding polymer. These results confirmed that the structures of the obtained resins were consistent with their putative structures [12]. The peak at 8.6 ppm was assigned to the protons of the phthalazinone moiety of the PPEKs molecule, near the carbonyl, which was H^10^ of PPEK, H^12^ of PPEK-M, and H^14^ of PPEK-Ph. In addition, there were no other peaks around 8.6 ppm. However, the chemical shift of the corresponding proton of the phthalazinone monomer was believed to appear at 8.3 ppm. It was indicated that the biphenyl (DHPZ, DHPZ-M, DHPZ-Ph) monomers were completely consumed in the nucleophilic substitution. It also shows that the product had been processed to be competent for all measurements. Furthermore, the structures of these polymers were further measured by FTIR, and the FTIR spectrum of the obtained polymer resins, as shown in Figure 1b. The infrared spectrum of the obtained resins were similar. The weak absorption peak around 3040 cm^-1^ was due to the stretching vibration of the unsaturated C-H on benzene. In addition, the strong absorption peaks appearing at 1592, 1,98, 1447, and 1480 cm^−1^ were attributed to the deformation vibration of the benzene ring. Among them, the absorption peak near 1,447 cm^−1^ was considered as the skeleton vibration of the benzene [17], which was due to the presence of conjugated substituents on the benzene, such as phenyl. Therefore, the intensity of the absorption peak of PPEK-Ph near the 1,447 cm^−1^ was significantly higher than that of the others. The wavenumber of the C=O and C–O bond absorption peaks in the PPEKs resin were 1,670 and 1,230 cm^−1^, respectively. The appearance of a strong absorption peak of the ether bond in the FTIR spectrum of all resins demonstrated that the nucleophilic substitution reaction was successfully carried out between the two monomers.

The morphological characteristics of PPEKs were identified by WAXD, and the results are shown in Appendix A. There was no doubt that all resins were amorphous, which was consistent with literature [18,19]. The broad reflection peak appearing around 2θ of 20° was noticed in the WAXD patterns of all PPEKs resins. The phthalazinone monomer must have been responsible for this because its unique structure destroyed the regularity of the molecular chain [12].

### 3.3. Thermal Properties

The thermal properties of PPEKs were investigated by DSC and TGA in different atmospheres. Figure 2a shows that the transition of PPEK, PPEK-M, and PPEK-Ph, from glass-state to rubber-state at 265, 269, and 255 °C, respectively. Furthermore, the *T_g_* value of the PPEK-M was higher than that of the PPEK, but the *T_g_* value of the PPEK-Ph was lower than that of the PPEK. By comparing the molecular chain structure of PPEK, PPEK-M, and PPEK-Ph, a possible mechanism was obtained and the relevant diagram is shown in Figure 2 (right). It is well-known that the molecular chains of PPEK resin containing the phthalazinone moiety, exhibit fully rigid and twisted characteristics. As a result of this, the molecular chains of PPEK resin were hard to move and had a high glass transition temperature. However, it must be admitted that there were still plenty of gaps within a molecular chain of PPEK resin. Parts of the gaps might have been filled by the methyl of the PPEK-M molecular chain. This made the movement of the molecular chains of PPEK-M more difficult than that of PPEK. Therefore, the *T_g_* value of the PPEK-M was higher than that of the PPEK. However, as the volume of the phenyl was larger than the methyl, rather than filling the gaps within a molecular chain, the phenyl of the PPEK-Ph, principally increased the space between the molecular chains. Thus, the entanglements of the molecular chains were reduced and the mobility of the molecular chains was enhanced, and the *T_g_* value of the PPEK-Ph was lower than that of the PPEK.

Computer simulation has been widely utilized as an auxiliary method to explain the experimental phenomena or to verify the validity of the mechanism [20,21]. In order to confirm the validity of the previous mechanism, an amorphous cell model of the PPEKs resin were successfully constructed by computer simulation, according to the literatures [13,22], and the patterns are illustrated in Figure 3. In order to preferably understand these graphs, some parameters such as the free volume and the fractional free volume (FFV) [23,24,25,26] of the resin were obtained, by using the Atom Volume and Surfaces Tool of MS 2017R2, and the parameters are shown in Table 2. Obviously, the value of the free volume of the PPEK-Ph (6681.68) molecular chain was the highest, followed by the PPEK-M (5915.62), and the PPEK (5737.64) was the lowest. It could be concluded that the introduction of phenyl into the phthalazinone moiety could effectively increase the free volume of the polymer, thus, resulting in a decreased Tg. In addition, an interesting phenomenon was captured, which was the simultaneous increase of the free volume and glass transition temperature of PPEK-M resin containing the phthalazinone moiety. However, this happened to be evidence of the previous speculation. Some methyl of the PPEK-M molecule existed in the gaps within the molecule and hindered the movement of the molecular chain, thus, resulting in an increased *T_g_*. The remaining methyl groups were arranged in the space between the molecular chains, which enhanced the free volume of the resin. In short, with the help of computer simulations, the truth was revealed.

The thermal stability and thermo-oxidative stability of the obtained resins were determined by TGA in different atmospheres (N_2_ and air). The TGA/DTG curves are displayed in Figure 4, while the corresponding data are shown in Table 3. The synthesized polymers were stable below 400 °C, in both nitrogen and air. It was observed that the polymers retained 95% of their weight up to 454–516 °C and 90% of their weight up to 504–530 °C, in N_2_. The thermal stability of PPEK and PPEK-Ph resin was equivalent, suggesting that the thermal stability of the resin did not reduce when introducing phenyl into the molecular chain. Nevertheless, PPEK-M resin presented the worst thermal stability with T_d_^5%^ of 454 and 485 °C in nitrogen and air, respectively. This indicated that the introduction of methyl could reduce the thermal stability of the PPEKs resin. It might be ascribe to the weaker thermal stability of methyl, compared to phenyl. In addition, all prepared resins underwent two thermal degradation processes in N_2_ and air, according to the DTG curves. In the nitrogen atmosphere, the two thermal degradation temperatures peak of PPEK and PPEK-Ph resins were observed at about 520 and 590 °C, while that of PPEK-M resin was observed at about 450 and 588 °C. Previous research works [27,28] have speculated that the first thermal degradation process of PPEK and PPEK-Ph resin appearing at 520 °C, might be due to the degradation of the rearrangement or crosslinking of the phthalazinone moieties. The second thermal degradation process of PPEK and PPEK-Ph resin was emerging at 590 °C, probably owing to the degradation of the carbonyl or ether bond [29,30]. As for the PPEK-M resin, the first thermal degradation process of PPEK-M might be caused by the degradation of methyl, while the second degradation process might be because of the degradation of carbonyl or ether bonds. Of course, the thermal degradation of resins is complex and requires further investigation.

In order to clarify the effect of phthalazinone’s side-group on the thermal degradation kinetics of the PPEKs resin, the thermal degradation properties of resins were studied in nitrogen at different heating rates. The Kissinger method [31] has been confirmed to be one of the simplest methods to obtain the degradation activation energy (*E_a_*), without being familiar with the degradation mechanism. Hence, Kissinger’s method was used to process experimental data, the Kissinger plots of the polymers are shown in Figure 5 and the relevant results are placed in Table 4. Obviously, the activation energy of the first degradation process (E_a1_ = 243 kJ·mol^−1^) and the second degradation process (E_a2_ = 361 kJ·mol^−1^) of the PPEK resin were determined to be the highest. This meant that the PPEK resin exhibited the best thermal stability. In addition, the E_a1_ values of PPEK-M and PPEK-Ph resins were 178 and 230 kJ·mol^−1^, respectively. It indicated that the introduction of methyl or phenyl into the PPEKs resin reduced the thermal stability of the resin. Among them, the thermal stability of PPEK-M resin was considered to be the worst, and this was consistent with the results of the TGA analysis. Furthermore, the E_a2_ values of PPEK-M and PPEK-Ph resins were 265 and 255 kJ·mol^−1^, which was significantly lower than that of the PPEK resin. It was further demonstrated that introducing the methyl or phenyl into the PPEKs resin could impair the thermal stability of the resin.

### 3.4. Dynamic Mechanical Properties

The dynamic thermomechanical properties of the PPEKs resin were performed by the DMA measurements (Mettler-Toledo, Zurich, Switzerland). The storage modulus (E′) and the loss tangent (tanδ) curves of the PPEKs, as a function of temperature, are displayed in Figure 6, and the relevant data are summarized in Table 5. The storage modulus of the materials, represents the deformation resistance of the resin. If a material is considered to be more rigid and less deformed, it should have a larger storage modulus. The dynamic mechanical scan showed E′ values (at 30 °C) of approximately 3,668, 3,461, and 3,436 MPa for PPEK, PPEK-M, and PPEK-Ph. The PPEK resin had the highest storage modulus, probably due to there being bountiful entanglements between the PPEK molecules. The storage modulus of PPEK-M resin had declined, which might be because a fraction of the methyl group on the resin increased the distance between the molecular chains while most of it filled the gaps within the molecular chains. As for PPEK-Ph resin, the phenyl of the PPEK-Ph molecular chain might have increased the distance between the molecular chain. This led to a reduction of the entanglement between the PPEK-Ph molecular, and the rigidity of the PPEK-Ph resin was lowered. Thus, PPEK-Ph resin should have had a relatively low storage modulus. In addition, the PPEK-Ph resin had the lowest storage modulus, which possibly was because the phenyl was better than the methyl, in improving the distance between the molecules. In any case, all prepared resins had excellent dynamic mechanical properties at 30 °C.

The *T_g_* of the resins obtained by DSC and DMA are shown in Table 5. A very significant feature was found, in that, the order of *T_g_*(tanδ) values for the acquired resins, differed from that of the *T_g_*(DSC) values. From the tan δ curves of the resin, the PPEK replaced the PPEK-M resin as the resin with the highest transition temperature. The *T_g_*(tanδ) values of the PPEK-M was higher than that of PPEK-Ph, and the *T_g_*(tanδ) values of the PPEK-Ph was the lowest. Probably due to the presence of thermal and force fields, the movement of methyl or phenyl of the molecular chain was enhanced. More methyl groups on the PPEK-M molecular chains were involved in expanding the distance between the chains rather than restricting the movement of the chains. Thus, the PPEK-M molecular chain could achieve free motion at a lower temperature. Phenyl on the PPEK-Ph resin was better at exploring the distance between molecular chains than methyl on PPEK-M resin. Therefore, the *T_g_*(tanδ) of the PPEK-Ph resin was the lowest. Nevertheless, all resins exhibited excellent thermomechanical properties that were not significantly reduced by the introduction of methyl or phenyl.

### 3.5. Rheological Properties

To investigate the processability of the prepared resins, rheological studies on the experimental resins were performed on a TA ARX2000 instrument (TA Instruments, Newcastle, DE, USA), at an oscillator frequency of 1 Hz from 280 to 420 °C. Figure 7 displays the melting viscosity as a function of the temperature for the prepared polymers. The melting viscosity of the PPEKs tended to slightly decrease, reached an equilibrium, and then increased as the temperature increased. When the sample was placed on the analysis platform, the melting viscosity of the resin at a specific temperature was received. At about 280 °C, the melting viscosity of the PPEK, PPEK-M, and PPEK-Ph were 1.26 × 10^5^, 7.08 × 10^4^, and 4.26 × 10^4^ Pa∙s, respectively. This might be attributed to the fact that the presence of methyl or phenyl could reduce the entanglement of the PPEKs molecular chains. Phenyl might produce more positive effects than methyl in reducing entanglement, so the melting viscosity gradually decreased. This assumption was consistent with the assumption of DMA. As the temperature raised, the entanglement of the intermolecular chains could be tardily broken by shearing, which resulted in a decrease in the melting viscosity. When the temperature reached 380 °C, the melting viscosity values of the PPEK-M had reached the lowest. Then, its viscosity raised rapidly. It was speculated that partial crosslinking or degradation of the PPEK-M resin might have occurred. As the temperature increased further, the viscosity values of the PPEK and PPEK-Ph resins came to their valleys, which were 1.50 × 10^4^ and 3.51 × 10^3^ Pa∙s. It was indicated that introducing the phenyl into the phthalazinone moiety could significantly improve the processability of the PPEKs resin. For methyl, it was difficult to achieve satisfactory results, due to the limitation of its thermal resistance.

### 3.6. Tensile Properties

The tensile properties of the resins were measured at room temperature, the stress–strain curve of the resin is shown in Appendix A. Relevant parameters, such as Young’s modulus (E_t_), tensile strength (σ_t_), and elongation at break (ε_t_) values of the polymers are presented in Table 6. Clearly, the E_t_ and σ_t_ values of the PPEKs resin increased with the addition of methyl and phenyl, but the fracture elongation of the polymers did not change significantly. In a tensile measurement, the samples underwent a process of deformation, expansion, and destruction. In this procedure, the cross-sectional area of the samples shrunk gradually. This led to the distance between the molecular chains to become smaller. In this case, methyl or phenyl might have become an obstacle to the relative movement of the molecular chain. This obstacle absorbed a part of the stress, thereby, increasing the E_t_ and σ_t_ values of the PPEKs resin. Overall, the tensile properties of PPEKs resin were improved due to the addition of methyl or phenyl.

## 4. Conclusions

The purpose of this study was to investigate the effects of phthalazinone’s side-group on the properties, especially the thermal stability and processability of the PPEKs resin containing phthalazinone moiety. When the PPEKs resin contained methyl or phenyl, the processability of the resin was significantly improved. This was because the presence of methyl or phenyl magnified the space between the molecular chains. Moreover, the mechanical properties of the PPEKs resin containing methyl or phenyl was also substantially enhanced. However, the PPEK-M resin could not achieve satisfactory results of improving the processability of the resin without reducing its thermal stability, due to the limitation of methyl’s thermal resistance. Therefore, it was considered to be a good method that introduced rigid group such as phenyl into the phthalazinone moiety, to enhance the processability of the PPEKs resin, while maintaining its thermal stability. In addition, it could also improve the mechanical properties of the PPEKs resin.

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
