# Peer review of "Study of the Effects of the Structure of Phthalazinone’s Side-Group on the Properties of the Poly(phthalazinone ether ketone)s Resins"

_polymers, 2019, doi:10.3390/polym11050803_

Reviewer 1 Report

This work was successfully conducted and the present manuscript is easy to read. There are some minor concerns to be published in POLYMERS.

1) For PEEK-M in Fig 1(a), please mark and explain what the peak at 7.25 ppm between No. 6 and 7.

2) It is better to modify the numbering of PEEK in Fig. 1(a). Following the numbering of PEEK, the new peaks in PEEK-M and PEEK-Ph can have new numbers. This is easier to compare different polymers.

3) In Fig. 1(b), the peak at 1447 cm-1 must be explained.

Author Response

Thank you very much for your kind suggestion and comments on our manuscript. We have revised the manuscript in line of them as follows.

Point 1: For PEEK-M in Fig 1(a), please mark and explain what the peak at 7.25 ppm between No. 6 and 7. 

Response 1: Thank you very much for the comments. The peak at 7.25 ppm is assigned to the proton of chloroform, which is a solvent. According to your advice, we have marked it in the Fig 1(a).

Point 2: It is better to modify the numbering of PEEK in Fig. 1(a). Following the numbering of PEEK, the new peaks in PEEK-M and PEEK-Ph can have new numbers. This is easier to compare different polymers. 

Response 2: Thanks for your suggestion. According to your suggesting, we have modified the numbering of protons of the PPEK, PPEK-M and PPEK-Ph in Fig. 1(a). Sure, it was easier to compare different polymers than before.

Point 3: In Fig. 1(b), the peak at 1447 cm-1 must be explained.

Response 3: Thanks for your comments. We must sincerely apologize for our carelessness to give you uncomfortable experience. According to your comments, the absorption peak at 1447 cm-1 in Fig. 1(b) is explained in lines 188~191. According to the reference (Reference 17), the absorption peak near 36175px-1 is considered as the skeleton vibration of the benzene, which is due to the presence of conjugated substituents on the benzene, such as phenyl.

Reviewer 2 Report

This paper reports the structural effects of Phthalazinone’s side-group on the properties of the poly(phthalazinone ether ketone)s resins. Characterizations on mechanical and thermal properties of PPEKs were appropriately carried out. Therefore, I recommend this manuscript to be accepted in Polymers journal.

Author Response

Thank you very much for your kind comments on our manuscript.

Reviewer 3 Report

The manuscript “Study of the Effects of the Structure of 3 Phthalazinone’s Side-group on the Properties of the 4 Poly(phthalazinone ether ketone)s Resins” presents the influence of side-groups of the phthalazione monomers on the polymer properties. It is well written and the findings are clearly presented in a structured manner.

Major comments:

1.       Characterization of synthesized DHPZ’s need to be improved:

-          DHPZ-M seems to be a new molecule, therefore it should be properly characterized: IR should be added also to prove the composition an Elemental analysis OR HRMs need to be performed.

-          For all the molecules NMR spectra – first, there are too many numbers after the comma for 13C-NMR. It should be one digit after the comma. Second, for all 1H- NMR there should be J-couplings reported in HZ, where multiplicity is stated (d, t, etc.).

-          You have a very detailed assignment of the carbon spectra, however there is no data how the assignment was performed (no additional experiments like HMBC or HSQC) – the assignment need to be discussed (at least in the SI)

2.       In the section 2.1 It is stated that you use CDCl3 for NMR , in the part where NMRs are discussed it is DMSO-d6 – please clarify.

3.       Section 2.2 Mw of the polystyrenes standard need to be stated, calibration curve should be added to the SI in the best way overlaid on the SEC chromatograms presented.

4.       Line 155-156: “In order to minimize the effects of molecular mass on the properties of the resin, the molecular mass of the resin was controlled to be 20000 g mol-1” It is stated that the Mw was controlled. How (time, solvent, etc.)? Or that was just a serendipity? If so, please rephrase.

Minor comments and misprints:

1.       Please, correct all significant figures (also known as the significant digits) of a numbers according to its measurement resolution.

2.       Section 2.2 please state the columns used for SEC (GPC)

3.       Take care of subscripts in your formulas: K2CO3

4.       Line 48: what does it mean “has total aryl”

5.       Line 84, 86, etc.: Use italic: 4-(4-Hydroxyphenyl) phthalazin-1(2H)-one, o-cresol

6.       Line 139: 100 mL –check through the document

7.       Line 161-162: “This can help to exclude the effect of molecular weight in a later discussion. were sufficiently high” Not clear – please correct.

8.       Line 290: being has; should be has

9.       What is pd Line 121 - if it is overlapping doublets report as multiplet, if you could resolve the system report as separate doublets (also these are likely not doublets).

Author Response

Thank you very much for the reviewer’s time and careful review. It is helpful for us to improve our manuscript. We have made the response point-by point as following:

Major comments:

Point 1: Characterization of synthesized DHPZ’s need to be improved:

Point 1(a): DHPZ-M seems to be a new molecule, therefore it should be properly characterized: IR should be added also to prove the composition an Elemental analysis OR HRMs need to be performed.

Response 1(a): Thank you very much for your advice. According to your suggestion, the structure of DHPZ-M was confirmed by using FTIR and HRMs, and the relevant results are shown in SI.

Fig. 1 The FTIR curve of DHPZ-M

Fig.2 The HRMs spectrum of DHPZ-M

Point 1(b): For all the molecules NMR spectra – first, there are too many numbers after the comma for 13C-NMR. It should be one digit after the comma. Second, for all 1H- NMR there should be J-couplings reported in HZ, where multiplicity is stated (d, t, etc.).

Response 1(b): Thank you very much for your advice. Based on your comments, we have revised the relevant statements in the manuscripts.

DHPZ-M: 1H NMR (400MHz, DMSO-d6, TMS):δ=12.79 (s, 1H, 8-NH), 9.75 (s, 1H, 1-OH), 8.47 – 8.29 (m, 1H, 7-H), 7.98 – 7.83 (m, 2H, 5&6-H), 7.81 – 7.75 (m, 1H, 4-H), 7.33 (t, J = 6.6 Hz, 1H, 9-H), 7.28 (dd, J = 8.2, 2.2 Hz, 1H, 3-H), 7.02 (d, J = 8.2 Hz, 1H, 2-H), 2.27 (s, 3H, 10-H)  13C NMR (400 MHz, DMSO, TMS) δ = 159.6 (s, 12-C), 156.6 (s, 1-C), 147.1 (s, 6-C), 133.9 (s, 8-C), 132.1 (s, 13-C), 131.8 (s, 9-C), 129.7 (s, 11-C), 128.4 (s, 3&5-C), 127.3 (s, 7-C), 126.5 (s, 4-C), 126.1 (s, 10-C), 124.5 (s, 14-C), 114.9 (s, 2-C), 16.5 (s, 15-C).

DHPZ-Ph: 1H NMR (400 MHz, DMSO-d6, TMS): δ = 12.84 (s, 1H, 8-H), 10.06 (s, 1H, 1-H), 8.37 (dd, J = 7.7, 1.3 Hz, 1H, 7-H), 7.98 ~ 7.79 (m, 3H, 5&6&9-H), 7.70 ~ 7.61 (m, 2H, 10-H), 7.50 (t, J = 5.3 Hz, 1H, 4-H), 7.48 ~ 7.40 (m, 3H, 3&11-H), 7.40 ~ 7.30 (m, 1H, 12-H), 7.18 (d, J = 8.3 Hz, 1H, 2-H). 13C NMR (400 MHz, DMSO, TMS) δ =159.7 (s, 12-C), 155.5 (s, 1-C), 146.8 (s, 6-C), 138.5 (s, 15-C), 134.0 (s, 13-C), 131.9 (s, 8-C), 131.8 (s, 7-C), 130.1 (s, 3-C), 129.7 (s, 11-C), 129.6 (s, 16-C), 128.5 (s, 17-C), 128.4 (s, 5-C), 128.2 (s, 14-C), 127.3 (s, 18-C), 127.2 (s, 9-C), 126.8 (s, 4-C), 126.5 (s, 10-C), 116.6 (s, 2-C).

Point 1(c): You have a very detailed assignment of the carbon spectra, however there is no data how the assignment was performed (no additional experiments like HMBC or HSQC) – the assignment needs to be discussed (at least in the SI).

Response 1(c): Thanks for your comments and we must apologize for our carelessness. According to your suggestions, the supplementary experiments (HMBC and HSQC) for DHPZ-M and DHPZ-Ph are conducted. Based on the results of experiments, the carbon spectrum was reassigned and the results of assignments were showed in SI.

Fig. 3 2D NMR spectrum of DHPZ-M and DHPZ-Ph: (a) is HMBC of DHPZ-M, (b) is HMBC of DHPZ-Ph, (c) is HSQC of DHPZ-M and (d) is HSQC of DHPZ-Ph

Point 2: In the section 2.1 It is stated that you use CDCl3 for NMR, in the part where NMRs are discussed it is DMSO-d6 – please clarify.

Response 2: Thanks for your suggestion. We must sincerely apologize for our carelessness to give you uncomfortable experience. According to your comments, “or DMSO” has been added in line 97. There are two deuterium substitution reagents used in this paper. One is chloroform for polymer testing while another is DMSO for monomer testing.

Point 3: Section 2.2 Mw of the polystyrenes standard need to be stated, calibration curve should be added to the SI in the best way overlaid on the SEC chromatograms presented.

Response 3: Thanks for your comments. According to your suggestion, we have superimposed the calibration curve and the SEC chromatograms together, and put it in SI. In addition, the Mw of the polystyrene was marked on the superimposed graph.

Fig.4 The GPC traces of PPEK, PPEK-M and PPEK-Ph.

Point 4: Line 155-156: “In order to minimize the effects of molecular mass on the properties of the resin, the molecular mass of the resin was controlled to be 20000 g mol-1” It is stated that the Mw was controlled. How (time, solvent, etc.)? Or that was just a serendipity? If so, please rephrase.

Response 4: Thanks for your comments. According to your comments, we have added " by adjusting the molar feeding ratio of monomer while maintaining the reaction temperature and time consistence " in line 165~166. First, the reaction degree of the monomer is assumed to be 1, and then the feed ratio of the monomer is calculated to obtain the polymer reaching the target molecular weight. At the same time, the reaction time and temperature of the resin are kept uniform during the polymerization. So, it is no serendipity that it is under our control.

Minor comments and misprints:

Point 1: Please, correct all significant figures (also known as the significant digits) of a number according to its measurement resolution.

Response 1: Thanks for your comments. According to your suggestion, we have corrected the significant figures of carbon spectrum and GPC.

Point 2: Section 2.2 please state the columns used for SEC (GPC)

Response 2: Thanks for your comments. The chromatographic columns used in this paper were composed of two PLgel MIXED-C (5 m x 300 mm) columns. We have added " And the chromatographic column was composed of two PLgel MIXED-C (5 m × 300 mm) columns " at line 100~101 in the manuscript.

Point 3: Take care of subscripts in your formulas: K2CO3

Response 3: Thanks for your comments. Based on your comments, we have checked and corrected the errors in subscripts such as AlCl3 and K2CO3 in the manuscript.

Point 4: Line 48: what does it mean “has total aryl”

Response 4: Thanks for your comments. We must sincerely apologize for the trouble caused by our carelessness. In line 48, we have changed the “has total aryl” to “has totally aromatic”. Here is based on that both benzene ring and heterocyclic ring containing in the phthalazinone moiety are aromatic, thus claiming that phthalazinone has a totally aromatic structure.

Point 5: Line 84, 86, etc.: Use italic: 4-(4-Hydroxyphenyl) phthalazin-1(2H)-one, o-cresol

Response 5: Thank you for your comments. According to your opinion, we have changed "o" to "o","2H" to "2H" in line 84&86.

Point 6: Line 139: 100 mL –check through the document

Response 6: Thank you for your comments. According to your opinion, we have changed "100mL" to "100 mL". In addition, we also changed "2h" to "2 h","12h" to "12 h".

Point 7: Line 161-162: “This can help to exclude the effect of molecular weight in a later discussion. were sufficiently high” Not clear – please correct.

Response 7: Thank you for your comments. According to your comments, we have deleted ". were sufficiently high ".

Point 8: Line 290: being has; should be has

Response 8: Thank you for your suggestion, and I have revised it based on your advice. Personally, I think "should has" might be better. I hope you can accept my modification.

Point 9: What is “pd” Line 121 - if it is overlapping doublets report as multiplet, if you could resolve the system report as separate doublets (also these are likely not doublets)

Response 9: Thank you for your comments. According to your suggestion, we have reprocessed the system report. And we found that the previous analysis was unreasonable. Thus, we changed the "pd" to "m" in line 122.
